# In-situ measurement of organic aerosol molecular markers in urban Hong Kong: temporal variations and source apportionment in summer

Hongyong Li[1,2], Xiaopu Lyu[2*], Likun Xue[1*], Yunxi Huo[3], Dawen Yao[4], Haoxian Lu[5], Hai Guo[3*]

[1]Environment Research Institute, Shandong University, Qingdao, 266237, China
[2]Department of Geography, Faculty of Social Sciences, Hong Kong Baptist University, Hong Kong
[3]Department of Civil and Environmental Engineering, The Hong Kong Polytechnic University, Hong Kong
[4]School of Intelligent Systems Engineering, Sun Yat-Sen University, Shenzhen, 518107, China
[5]Marine Biological Resources Bank, Southern Marine Science and Engineering Guangdong Laboratory (Zhuhai), Guangdong, China

*Correspondence to*: Xiaopu Lyu (xiaopu_lyu@hkbu.edu.hk); Likun Xue (xuelikun@sdu.edu.cn); Hai Guo (hai.guo@polyu.edu.hk)

**Abstract.** Organic aerosol (OA) is a significant constituent of urban particulate matter (PM), and molecular markers therein provide information on the sources and formation mechanisms. With in-situ measurement of over 70 OA molecular markers in a summer month at a bihourly resolution, this study focused on the temporal variations of representative markers and dynamic source contributions to OA at an urban site in Hong Kong (HK). The levels of secondary OA (SOA) markers were markedly elevated ($p < 0.05$) in the short-duration cases with continental and coastal air relative to those in the most common marine air, and the primary markers were more of local characteristics. The troughs witnessed the enhancements of many SOA markers ($p < 0.05$), which appeared to be related to the high relative humidity. The diurnal patterns of 2-methyltetrols differed between scenarios, and their aqueous formation at night seemed plausible, particularly in the presence of troughs. Eight unambiguous sources were identified for the organic matters in submicron PM ($PM_1$-OM). Despite an urban site, the mean SOA contribution ($66.1 \pm 10.5\%$), primarily anthropogenic, was significant. Anthropogenic SOA dominated in the cases with continental and coastal air and in early afternoon. Local cooking and vehicle emissions became predominant in the scenario of marine air without troughs. Even averaged over the study period in this summer month with the prevalence of marine air, primary cooking emissions contributed up to 44.2% to $PM_1$-OM in the early evening. The study highlighted the need to control regional anthropogenic SOA and local cooking emissions to mitigate PM pollution in HK.

**Keywords:** Organic aerosol; Molecular marker; Source apportionment; Cooking emissions; Anthropogenic secondary organic aerosol

## 1 Introduction

Atmospheric particulate matter (PM) has been a widespread environmental concern, due to its significant climate and health effects (Ramanathan et al., 2001; Poschl, 2005; Yang et al., 2018). Organic aerosols (OA) make up 20-90% of fine PM ($PM_{2.5}$)

by mass concentration (Kanakidou et al., 2005; Jimenez et al., 2009). While the inorganic PM compositions are somewhat being effectively controlled, reducing OA remains a major challenge for further mitigation of $PM_{2.5}$ pollution in China (Huang et al., 2014; Zhang et al., 2019). This is closely related to the lag and inadequacy of our knowledge of OA, especially the

insufficient understanding of time-resolved and unambiguous identification of OA sources and SOA formation mechanisms. Therefore, it becomes increasingly important to improve the understanding of OA with new theories and techniques.

Molecular information records history of OA, including sources and chemical evolutions. Filter-based technique is a widely used method for measuring molecular compositions of OA (Yu et al., 2011; Ding et al., 2015; Fu et al., 2016). It generally takes half a day to a few days to collect PM on filters, followed by solvent or thermal extraction of target compounds and

chemical analysis. However, during the collection cycle of each sample, emissions of OA and its precursors are not necessarily consistent, and chemical transformation may occur over several generations. For example, cooking and vehicle emissions vary significantly over the course of a day (Sun et al., 2018; Yao et al., 2021). With much higher time resolutions, Aerosol Mass Spectrometry (AMS) was developed and has been extensively applied in OA research (Sun et al., 2018; Liu et al., 2019; Yao et al., 2022). Nevertheless, AMS cannot provide molecular information of OA due to the application of hard ionization, which

makes it difficult to identify unambiguous OA sources with AMS data (Sun et al., 2011; Lee et al., 2013). Applied earlier in the United States (Williams et al., 2006, 2010), in-situ measurement of OA molecular markers based on thermal desorption has become an emerging technique for chemical characterization of OA in China, *e.g.,* Beijing (Ren et al., 2019), Shanghai (Li et al., 2020; Wang et al., 2020), and HK (Lyu et al., 2020; Wang et al., 2022).

HK is a metropolis in East Asia with a developed economy and a high population density. Although the $PM_{2.5}$ level continued

to decrease in recent decades, it is still far from the World Health Organization guideline value (5 μg m$^{-3}$). Led by the tertiary industry, HK has few high-emission industries. Street-level air pollution has long been a concern due to vehicle emissions (Lee et al., 2006; Huang et al., 2014; Liu et al., 2017a). Besides, mounting evidence demonstrates the considerable contributions of cooking emissions to OA (Lee et al., 2015; Sun et al., 2016). On the other hand, SOA formation in HK may also be significant, as mediated by the subtropical climate and coastal features. For example, studies indicated that aqueous processes accelerated

SOA formation in spring with high relative humidity in HK (Li et al., 2013). In addition to local emissions and chemistry, transboundary transport also increases the atmospheric load of OA in HK (Huang et al., 2014; Lyu et al., 2017). With filter-based techniques, a range of molecular compositions of OA, such as polycyclic aromatic hydrocarbons (PAHs) (Guo et al., 2003; Liao et al., 2020), alkanes (Yao et al., 2004; Yu et al., 2011), dicarboxylic acids Ho et al., 2006; Hu et al., 2013), and anhydrosugars (Zhang et al., 2012; Ho et al., 2014), have been well studied in HK. Further, OA sources have also been explored

by introducing the molecular markers into receptor models (Hu et al., 2010; Huang et al., 2014; Chow et al., 2022). Cooking emissions, vehicle emissions, biomass burning, and secondary formation were identified as the main sources of PM in HK. AMS has also been applied in HK, advancing our understanding of the dynamic features of OA and its sources (Lee et al., 2015; Xing et al., 2022). In recent years, Thermal desorption Aerosol Gas chromatography coupled with mass spectrometry (TAG) allowed in-situ measurement of OA molecular markers and facilitated understanding of dynamic source contributions

to OA at two background sites in HK (Lyu et al., 2020; Wang et al., 2022). In brief, it was found that the concentrations of

SOA tracers were significantly elevated in PM$_{2.5}$ pollution episodes, and transboundary transport led to a rise in biomass burning contributions to OA. Despite the previous studies, there is a lack of knowledge on time-resolved and unambiguous sources of OA in urban areas of HK.

This study presents the temporal variations of OA and a range of molecular markers therein based on a continuous measurement for ~1 month at an urban site in HK with TAG and AMS. Diurnal patterns of typical markers are discussed to indicate the emission patterns and possible formation mechanisms of OA. Source apportionment with the time-resolved data reveals the dynamic source contributions. The study enhances our understanding of urban OA in HK at bihourly resolution and molecular level, and provides a reference for PM pollution mitigation by intervention in OA emissions and formation.

## 2 Methods

### 2.1 Sampling campaign

The observation experiment was carried out in the campus of the Hong Kong Polytechnic University from June 6th to 26th in 2019. The instruments were deployed in a laboratory on 11th floor with a height of ~45 m above ground level. The sampling inlet protruded ~1.5 m from the wall. As shown in Figure 1, the site was in a mixed residential and commercial area with intensive traffic, which was also one of the areas with the highest concentration of restaurants in HK. Besides, it is worth noting that the nearby funeral parlors and temples within 1 km to the east of the site often conducted the burning of worship supplies, ritual candles, and incenses.

Despite an urban site, air pollution could also be influenced by regional transport. The 72-h backward trajectories of air masses reaching the site were calculated using the Hybrid Single Particle Lagrangian Integrated Trajectory model, which was driven by the Global Data Assimilation System archive data at the horizontal resolution of 1°×1°. The trajectories started from the location 50 m above the sampling site and were calculated every 2 hours. Based on their origins and paths, the air masses were classified into three types (Figure 1). Those from the South China Sea represented the marine air (blue trajectories in Figure 1) and accounted for ~87% of the total air masses. However, the fraction of air masses originating from and passing over the mainland was very low (6%, red trajectories), same for those arrived at the site along the coastline of eastern and southern China (7%, orange trajectories). From a representative perspective, we discuss them as special cases, and the continental air and coastal air are labelled as Case I and Case II, respectively.

To analyze OA in small particles of interest, a submicron particulate matter (PM$_1$) cyclone was installed at the front of the inlet line made of pre-cleaned copper tubing. The cut-off size of 1μm also matched the design of the High Resolution − Time of Flight − Aerosol Mass Spectrometer (HR-Tof-AMS, referred to as AMS hereinafter) manufactured by Aerodyne Research Inc. The AMS couples size-resolved particle sampling and mass spectrometry analysis, enabling real-time measurement of non-refractory compositions of PM$_1$, including total organic matter (PM$_1$-OM), sulfate, nitrate, ammonium, and chloride. With hard ionization and in absence of species separation, AMS only provides mass spectral fragmentation information of size-segregated particles that arrive at the ionizer at the same time. As an important addition, the molecular markers of PM$_1$-OM

were measured every 2 hours using the first commercialized TAG jointly developed by Aerodyne Research Inc and University of California, Berkeley. The main difference between TAG and traditional gas chromatography – mass spectrometry (GC-MS) is that TAG automates sample collection, derivatization, transfer, and GC-MS analysis. Within each cycle, the air after the removal of gaseous organics by a charcoal denuder was drawn and impacted onto a collection and thermal desorption (CTD) unit at a constant rate of ~10 L/min in the first 90 minutes. Afterwards, the CTD was heated according to a preset program. At the same time, a helium flow saturated with N-Trimethylsilyl-N-methyl trifluoroacetamide was injected into the CTD to derivatize and purge the sample to a focusing trap (FT) that was maintained at room temperature. The FT allowed the volatiles to flow out and trapped the less-volatile compounds. Then, it was heated quickly, and the trapped substances were flushed to the chromatographic column head by helium. Lastly, the GC-MS analysis was performed, which however did not take any additional time, because it happened simultaneous with sample collection. The sample treatment, transfer, and some other steps took 30 minutes, thereby 2 hours for a complete cycle. With online derivatization and thermal desorption, the TAG was capable of analyzing a wide range of polar and nonpolar organic compounds. A 5.22 μL of mixture of deuterated ($^2$H-containing) or $^{13}$C-containing compounds at the concentration of 0.0625-0.5 ng/μL was used as internal standards (IS), which was injected on top of every ambient sample and subject to the same treatment and analysis procedures. Therefore, the ISs could track and correct for the changes in instrument sensitivity. More details about the AMS and TAG operations, including the IS information, can be found in our previous studies (Lyu et al., 2020; Huo et al., 2022).

Besides, the mixing ratios of inorganic trace gases, including sulfur dioxide ($SO_2$), carbon monoxide (CO), nitric oxide (NO), nitrogen dioxide ($NO_2$), and ozone ($O_3$) were continuously measured in the sampling period. The instruments are listed in Table S1, and more operation and quality control details have been presented elsewhere (Guo et al., 2013). We adopted the meteorological parameters measured by the Hong Kong Observatory at the King's Park Meteorological Station, which was ~870 m away from our sampling site. They include temperature (Temp), relative humidity (RH), wind speed (WS), wind direction (WD), and ultraviolet (UV). The prevailing winds were roughly consistent with the origins of air masses indicated by the backward trajectories. Liquid water content (LWC) in $PM_1$ was calculated in the same way as that adopted by Yao et al. (2022), using the Extended Aerosol Inorganics Model with Temp, RH, and inorganic ions in $PM_1$ as inputs. Table S2 shows the various metrics (mean, median and percentiles) of the trace gases, meteorological parameters and LWC.

## 2.2 Processing of TAG data

A total of 225 bi-hourly samples were analyzed by TAG throughout the experiment, and 74 OA markers in the volatility range equivalent to $C_{16}$ – $C_{36}$ n-alkanes were identified based on their retention time and mass spectrums, especially the base ion (Lyu et al., 2020), as shown in Table S3. In brief, we identified a species from the TAG chromatogram based on the retention time of the base ion of authentic standard and confirmed it by searching the ion fragments in an expanded NIST mass spectral library. For a few samples, the fatty acids and oleic acid were not fully derivatized, likely due to their exceptionally high concentrations in cooking hours. Peak fitting and integration were performed to both the derivatized and underivatized forms of them. Given the consistent amount of ISs injected on top of all individual samples, the variations in peak areas of ISs over

time could only be attributed to changes in instrument sensitivity (including recovery rate). To understand the true variations of target compounds in the air, we calculated the ratio of the peak areas of target compounds to the peak areas of corresponding ISs with similar structures and close retention time, and the ratio is deemed as IS-corrected response. The pairing of target compounds and ISs are the same as that adopted in Lyu et al. (2020). However, we ran out of the external standards (multiple concentrations of authentic standards that are exactly the same as target compounds or surrogate standards) in this sampling campaign. Therefore, the relationship between IS-corrected response and concentration was not determined, and the concentrations of the detected species were not quantified. We use the IS-corrected responses of target compounds for analysis instead.

For ease of analysis, we first classified the detected compounds based on their common origins and chemical structures. Further, Pearson correlation analysis was performed among them, as shown in Figure 2. Moderate to good correlations were identified for some species within 7 groups, including common SOA tracers, dicarboxylic acids (DCA) & hydroxyl dicarboxylic acids (OHDCA), sugars & sugar alcohols, *n*-alkanes, fatty acids, aromatic acids, and PAHs. The Pearson correlation coefficient (r) was higher than 0.89 for the correlations between any pairs of 4-/5-ring PAHs. The variations of levoglucosan and mannosan were roughly consistent (r = 0.83). Good correlations (r > 0.78) were identified within the subgroups of high molecular weight ($C_{25} - C_{30}$) and low molecular weight ($C_{20} - C_{24}$) *n*-alkanes, separately, but not between them. This likely resulted from the different sources of them, *e.g.,* vegetable wax and debris for the former and petroleum products for the latter (Kawamura et al., 2003; Simoneit et al., 2004; Kang et al., 2016a). Consistent variations were observed for OHDCA species. Some DCAs, such as succinic acid which is a known precursor of malic acid (Hu et al., 2013), also correlated well with OHDCA. The patterns of isoprene SOA tracers formed under the conditions of low nitrogen oxides ($NO_x$), *i.e.,* 2-methyltetrols (2-MTs) and $C_5$-alkenetriols (Claeys et al., 2004; Edney et al., 2005; Surratt et al., 2010), resembled each other (r > 0.88). Besides, there also existed some correlations between species of different groups, mainly due to co-emissions from cooking (oleic acid, fatty acids, fructose isomers, and azelaic acid) and biomass burning (vanillic acid (VA), 4-hydroxybenzoic acid (4-OHBA), and levoglucosan). Based on the correlations, we selected 18 OA markers as representatives of the whole in the following analyses, as highlighted in Figure 2 and Table S3. All the chemical and meteorological data with time resolution higher than TAG data were converted to bihourly averages for matching purposes where necessary.

## 2.3 Application of Positive Matrix Factorization (PMF) model

PMF has been widely used as a quantitative tool to understand air pollution sources (Hu et al., 2010; Wang et al., 2018; Dai et al., 2020). In this study, the EPA PMF v5.0 was used to perform the source apportionment for $PM_1$-OM. A total of 13 OA markers measured by TAG and $NO_x$ were used to indicate the sources, including 2 OHDCA species (MA and 2-HGA, see Table S3 for the full names), 2 anthropogenic SOA tracers (DHOPA and phthalic acid), 3 isoprene SOA tracers (2-MT1, cis-2-MBT, and 2-MGA), 1 monoterpenes SOA tracer (HDMGA), 1 oxygenated cooking OA (OCOA) tracer (9-oxononanoic acid), and 4 primary organic aerosol (POA) species (levoglucosan, fructose-2, palmitic acid, and oleic acid). The uncertainty file was compiled following the method described in Lyu et al. (2020), based on an error faction of 10% and method detection

limit (MDL) for individual species. The MDL for $PM_1$-OM measured by AMS was determined to be 0.201 μg/m$^3$, as described in Text S1. Since the molecular markers measured by TAG were not quantified in this sampling campaign, the MDLs determined for the same instrument ~8 months earlier were adopted (Lyu et al., 2020). For the newly detected species, *e.g.,* phthalic acid, 2-MGA, cis-2-MBT, and 9-oxononanoic acid, we assume 10% of the average IS-corrected response as the dimensionless MDL. Extreme values are generally difficult to be reproduced by PMF model, especially for data points with scaled residuals greater than 4. To address this, greater weight was given to the high values that are considered true in the PMF iteration process, as described by Wang et al. (2018).

The optimal number of PMF factors was determined according to the mathematical tests and physical interpretability. We examined the solutions with 3 – 10 factors and the corresponding Q/Qexp ratios (Figure S1). The ratio decreased rapidly for 3 – 7 factors and the decline slowed down for 7 – 10 factors, demonstrating a minimum of 7 factors from a mathematical perspective. Further inspections found that some tracers indicative of different sources (*e.g.,* DHOPA and MA) were not separated in the 7-factor solution, and some species were not well reproduced. The 9-factor solution however apportioned tracers of a same source into multiple factors. We ended up with 8 factors, which provided reasonably good fit for all species with physically interpretable source profiles.

## 2.4 Statistical analysis

Due to uneven emission patterns, non-linear chemistry and changing meteorological impacts, the distribution of urban air pollutants concentration is often uncertain. This determines how we describe the data. In this study, we examined the data distribution with the Shapiro-Wilk tests. It was found that the concentrations of most species studied did not follow a normal distribution, neither over the entire observation period nor within sub-periods defined by air mass categories, with only a few exceptions (e.g., palmitic acid in Case I and 2-MGA in Case II). Therefore, it is insufficient to analyze and discuss the data based only on the mean values. As a supplement, we also show the 25$^{th}$ percentile, median and 75$^{th}$ percentile where necessary. However, to avoid redundancy, not all the metrics are referenced in all discussions. Moreover, given the non-normal distribution of the data, non-parametric Mann-Whitney U tests were performed to compare any two sets of data. The significance levels are expressed as *p* values. Both the Shapiro-Wilk test and the Mann-Whitney U test were implemented using the scipy.stats package in Python. All the correlation coefficients shown in this paper are derived from Pearson correlation analysis and are significant ($p < 0.05$), unless otherwise specified.

## 3 Results and Discussion

### 3.1 Overview of the sampling campaign

Figure 3 and Figure S2 show the time series of the selected OA markers, $PM_1$ compositions, trace gases, and meteorological conditions throughout the sampling period. The origins of air masses are also labelled. Table S4 lists the statistics of the $PM_1$-OM, sulfate, nitrate, and ammonium in $PM_1$. $PM_1$-OM was found to be the most abundant component, followed by sulfate,

ammonium and nitrate, whether based on the mean or median. The fraction of $PM_1$-OM ($47.1 \pm 2.2\%$) was comparable to that at an urban background site in HK (43.8%), but significantly lower ($p < 0.05$) than that at a roadside site ~350 m away (57.7%) (Yao et al., 2021,2022). From the perspective of air mass categories, $PM_1$-OM was the lowest in the most common marine air, followed by that in Case II with coastal air and Case I with continental air, same for CO and $O_3$ (Tables S5-S7). The highest concentrations of sulfate and ammonium were observed in Case II, likely due to aqueous aging of regional air pollution (Huo et al., 2024). Although the marine air was supposed to be relatively clean, it contained elevated levels of $SO_2$ and $NO_x$, which correlated well with each other (r = 0.87). This phenomenon was in line with that at the three roadside sites in HK, but was not identified at the general urban sites (Figure S3). Therefore, the close relationship between $SO_2$ and $NO_x$ on the days with marine air were likely attributable to local vehicle emissions rather than ship emissions. For all the types of air masses, we also note that the variation of $NO_x$ throughout the day was highly consistent with that observed at a nearby roadside site (Figure S4). Moreover, there was no correlation between $NO_x$ and levoglucosan, even during the period with elevated levels of levoglucosan, eliminating the likelihood of high biomass burning contributions to $NO_x$. Therefore, $NO_x$ is regarded as a tracer of vehicle emissions in this study. Under the marine air conditions, the concentrations of many air pollutants, including both primary and secondary species, increased significantly ($p < 0.05$) on June 10th – 14th and 24th – 26th, when HK was affected by the troughs of low pressure with cloudy weather and low wind speeds. This might be related to the unfavorable dispersion conditions and enhanced chemistry under special meteorological conditions and is further discussed below.

The primary cooking OA (PCOA) markers, such as oleic acid, palmitic acid, and fructose isomers, did not show obvious inclination toward the trough periods and exhibited some periodic spikes in the marine air. Therefore, it seemed that local cooking emissions somewhat overwhelmed the effects of atmospheric mixing and transformation. However, the levels of PCOA markers were much lower ($p < 0.05$) in the case with continental air than in the marine air. On the one hand, this could be partially explained by the quicker chemical losses, due to the higher concentrations of oxidants, $e.g.,$ $O_3$ (Tables S5-S7). Oleic acid with a carbon-carbon double bond in the molecule can be efficiently oxidized by $O_3$ and other oxidants (Zeng et al., 2020; Wang et al., 2021). The argument is supported by the fact that 9-oxononanoic acid, a typical OCOA tracer (Lyu et al., 2021), in the two cases was 1.47-2.00 times (or 1.36-1.53 times by median) the level in the marine air. On the other hand, the differences could be related to the uneven distribution of restaurants on different air paths. As shown in Figure 1, the restaurant density in the areas where the continental and coastal airflows passed is substantially lower than south of the sampling site. Besides, it cannot be ruled out that the limited number of samples in the two cases coincidently did not capture the cooking emissions well. Differently, levoglucosan and some odd $n$-alkanes ($e.g.,$ $n$-$C_{27}$) with a few spikes were more abundant in Case II ($p < 0.05$). Further, we found that high levels of levoglucosan were associated with easterly winds (Figure S5). As stated in section 2.1, the burning of incense, candles, banknotes, paper products was common within 1 km east of the site. There was no known source of biomass burning further east, and east of HK are waters. Therefore, the observed variations of levoglucosan were mainly attributed to the combined effect of local emissions and wind patterns, different from the regional transport of biomass burning plumes from mainland to HK in cool seasons (Sang et al., 2011; Chow et al., 2015). The wind dependence

was not identified for the other species, except that the strong southwest winds caused low levels of many OA markers due to the dilution by the relatively clean marine air.

The temporal variations of SOA markers were quite different. The mean (median) values of 2-MTs (*e.g.,* 2-MT1), $C_5$-alkenetriols (*e.g., cis*-2-MBT), and 2-MGA increased significantly ($p < 0.05$) by 13.1 (7.3), 18.4 (19.5), and 13.8 (9.8) times in Case I when the continental air arrived. Consistent with that at an urban background site (Lyu et al., 2020), higher levels of 2-MTs (isoprene SOA in $NO_x$-lean environment) were observed at night and 2-MGA (isoprene SOA in $NO_x$-rich environment) exhibited the peak in the afternoon. The temperature and UV were not exclusively high in Case I, and we presume the same

for biogenic emissions locally. Therefore, the notable levels of isoprene SOA tracers suggested that the continental air transported biogenic SOA and/or the precursors to this site. Succinic acid correlated fairly well with 2-MGA in Case I (r = 0.92), indicating that it was a degradation product of isoprene under high-$NO_x$ conditions. Besides, the first day of Case I also experienced the highest levels of DHOPA and phthalic acid, two anthropogenic SOA tracers derived from aromatics (Bunce et al., 1997; Jang et al., 1997; Kleindienst et al., 2004). This could also be related to regional transport, given the notable rise

in other anthropogenic air pollutants (*e.g.,* CO) simultaneously. In addition, it was likely that the enhancement of atmospheric oxidation capacity, as indicated by the highest level of $O_3$ over the sampling campaign, facilitated the formation of many SOA species on this day. These factors might also play a role in maintaining the high levels of OHDCA, such as malic acid and tartaric acid, which were elevated during a low-pressure trough in the previous days with marine air. In fact, the troughs witnessed enhancements of a wide range of species relative to their levels in the marine air without troughs (Table S7). The

highest change ratio for the mean values of POA markers was 5.3 for benzo[a]pyrene. However, more significant increases were observed for many SOA markers, including DHOPA, $C_5$-alkenetriols, OHDCA species, and phthalic acid, implying the role of processes other than transport and atmospheric mixing. The conclusion also holds, based on median analysis. It is noteworthy that the troughs featured lower temperature and much weaker radiation, while the RH and LWC were higher (Table S7). A recent study demonstrated the aqueous photochemical formation of OHDCA in HK (Huo et al., 2024). While it is

unknown whether this was also responsible for the increases of the other SOA markers, a moderate correlation (r = 0.64) was identified between phthalic acid and the product of RH and UV during the trough periods and was not found at other times (Figure S6). Besides, the rise in $C_5$-alkenetriols resembled the changes in 2-MTs, which mainly occurred at night and could also be linked to aqueous processes, as discussed below. Therefore, the high RH and LWC might be one of the leading factors, if not all, of the SOA enhancements in the presence of troughs.

**3.2 Diurnal patterns of representative OA markers**

An advantage of TAG measurement is the time-resolved data of OA markers. Figure 4 presents the average diurnal profiles of 9 representative species, and some others with lower priorities are shown in Figures S7-S8. While both the mean and median are used in the following discussions, the much lower median values, if applicable, were generally observed in the marine air, and the high mean values resulted from a small number of samples influenced by continental air, coastal air or intense local

emissions.

The consistently higher median values for levoglucosan between 6:00 and 22:00 indicated its emissions from human activities. As discussed above, the elevated levels of levoglucosan were related to the combustion of biomass products nearby. Here, we note that the sporadic extreme high values caused a marked increase in mean values of levoglucosan during 18:00 – 20:00, which roughly coincided with the timing of smokes from the funeral parlor chimneys, according to our observations. Some other species, including 4-OHBA, VA, mannosan, xylose, hydroxybutyric acid, and protocatechuic acid showed consistent diurnal patterns (Figures S7-S8), indicating that they were mainly derived from biomass burning. Besides, both the bihourly mean and median $C_{26} - C_{31}$ $n$-alkanes were observed with obvious peaks at 8:00 and 20:00 and might also be related to the burning of incenses, candles, joss papers and paper artifacts nearby, which was confirmed in other scenes (Lyu et al., 2021; Song et al., 2023). Other sources, $e.g.,$ vehicle emissions, might also contribute to the peak at 8:00 that was more pronounced than levoglucosan. Oleic acid was obviously enhanced at noon and in early evening, when intensive cooking activities took place in restaurants and at home, same for fatty acids, azelaic acid, and fructoses (Figures S7-S8). The consistency of azelaic acid with oleic acid ($r = 0.76$) indicated that a considerable fraction of the azelaic acid was derived from or quickly formed in primary cooking emissions. Pyrene, as a representative of PAHs, was observed with higher median and mean values during 6:00-22:00. The median values showed peaks in the morning and early evening, while the mean value was the highest at 20:00. Interestingly, the high levels of pyrene correlated well with levoglucosan ($r = 0.94$) at 20:00 (Figure S9) and exhibited no correlation with oleic acid which also had the highest levels in the early evening. Therefore, the burning of biomass products nearby was responsible for the occasionally elevated levels of PAHs in the early evening. At the other times, the variation of pyrene was consistent with that of $NO_x$ ($r = 0.80$), indicating their common source of vehicle emissions. This also explained the bimodal distribution of the bihourly median values of pyrene.

As a tracer of OCOA, 9-oxononanoic acid got high levels at night and showed a peak in the early afternoon. The nighttime enhancement was in line with the observations in an indoor air quality study (Lyu et al., 2021), and was due to the $O_3$-initiated oxidation of cooking emissions that occurred in the early evening. For the same reason, the bihourly median 9-oxononanoic acid peaked at 14:00, 2 hours after the peak of oleic acid. Although the diurnal pattern of 9-oxononanoic acid was totally different from that of oleic acid and azelaic acid, the variation in the ratio of 9-oxononanoic acid to oleic acid was consistent with that in the azelaic acid to oleic acid ratio (Figure S10), both reflecting the chemical aging of cooking emissions. The bihourly mean 2-MTs showed higher levels between midnight and dawn, as did $C_5$-alkenetriols, contrary to the common understanding. This however was in line with previous observations at two background sites in HK conducted by us and another team (Lyu et al., 2020; Wang et al., 2022). Nevertheless, a significant daytime enhancement was observed for the bihourly median 2-MTs. Day-by-day inspections found inconsistent diurnal patterns for 2-MTs, which were grouped into four categories, as shown in Figure 5. As speculated in previous studies (Lyu et al., 2020), the higher levels of 2-MTs in the two cases with non-marine air might result from transboundary transport and diurnal evolution of boundary layer. On the days with marine air, two distinct patterns were identified. Similar night time enhancement was observed in the presence of troughs. Interestingly, 2-MTs exhibited the highest levels in the afternoon without the influence of the troughs, which resulted in the different patterns of bihourly mean and median values. These days were accompanied by higher temperature and UV intensity,

but lower RH and LWC (Figure 5). Therefore, the afternoon peak was likely attributed to the more intensive emissions and photochemical oxidation of isoprene. Moreover, we uncovered some clues that nocturnal chemistry might regulate the diurnal patterns of 2-MTs. According to Noziere et al. (2011), the ratio of 2-methylerythritol to the sum of 2-methylerythritol and 2-methylthreitol, referred to as 2-MT1/2-MTs hereinafter, indicates the sources of 2-MTs: primary emissions (0.35), $NO_x$-lean photooxidation (0.61), $NO_x$-rich photooxidation (0.76), and aqueous phase oxidation (0.90). Under the assumption that the

instrument sensitivity to the two 2-MT isomers were the same, the average 2-MT1/2-MTs ratio was 0.70 in the marine air without troughs. However, they were much higher (0.81) in presence of troughs, especially at night, in line with the patterns of LWC. Hence, the higher levels of 2-MTs at night during the trough periods might be related to aqueous chemistry. Nevertheless, it could not be the sole reason for the counterintuitive patterns of 2-MTs in the continental and coastal cases, where the average 2-MT1/2-MTs ratio was 0.76 and 0.77, respectively. The influence of other factors, such as transport and

atmospheric mixing, deserves study.

Conversely, 2-MGA exhibited a typical diurnal pattern of photochemical pollution, with the bihourly mean and median values increasing from morning to afternoon. Daytime enhancements were also observed for the mean values of DHOPA and phthalic acid, although their bihourly median were also high in some nighttime hours. The results suggest that photochemistry was involved in the formation of these SOA tracers. It is noteworthy that the peak of bihourly mean phthalic acid appeared at 12:00,

~4 hours earlier than the peaks of 2-MGA and DHOPA, which might be due to differences in formation mechanisms and/or oxidation states. For example, Yao et al. (2022) found that the peak of more oxidized oxygenated OA was several hours later than that of less oxidized OA, and aqueous processes were suspected between the two peaks. Despite a slight rebound at 12:00 – 14:00, the diurnal variation of bihourly mean malic acid was small, much less pronounced than that at an urban background site (Lyu et al., 2020). This pattern also applied to the other OHDCA species, which might be partially attributed to the lower

atmospheric oxidation capacity, as reflected by the difference in odd oxygen ($O_x = O_3 + NO_2$) levels, *i.e.,* 29.5 ppbv at this site *vs.* 49.8 ppbv at the background site. In contrast, the bihourly median value increased from 18:00 to 8:00 and got another peak at 14:00. These much lower levels of malic acid were observed in marine air, and the diurnal pattern was likely a combination of sea-land breezes (increase at night) and photochemical formation (afternoon peak).

### 3.3 Source apportionment of $PM_1$-OM

Figure 6 shows the source profiles of the 8 factors resolved by PMF. Levoglucoan was assigned to multiple factors in the base run, although the factor of biomass burning accounted for the highest fraction. Constraint was applied to reduce the levoglucosan loadings in other factors, thereby lower rotational ambiguity. Similarly, the loading of 9-oxononanoic acid in factors other than OCOA was also pulled down. Moreover, we also performed simulations without $PM_1$-OM that was measured by AMS. This was to examine if it was appropriate to mix the components measured by two different instruments in the source

apportionment. As shown, the source profiles did not change much between these simulations. Table S8 summarizes the r, slope, and intercept for the linear regression between the simulated and observed values of individual species, which are in reasonable ranges. The scaled residuals for all the species were within the range of -3 to 3. These metrics indicated that the

observed magnitudes and variations of the PMF species were well reproduced. To evaluate the model stability and uncertainty of the results, Bootstrapping (BS) and displacement (DISP) analyses were conducted. Based on BS tests, the factor mappings ranged from 92% to 100%, above the threshold of 80% representing robust factors recommended by the PMF user guide. Namely, there was no observation that disproportionately influenced the solution. Besides, no factor swap was found from the DISP analysis. The BS and DISP results indicated the robustness of the 8-factor solution.

The 8 factors were identified as PCOA, vehicle emissions, biomass burning, isoprene SOA (low $NO_x$), anthropogenic SOA represented by phthalic acid (ASOA-1), anthropogenic SOA represented by DHOPA (ASOA-2), OHDCA-like SOA, and OCOA. Factor 1 was dominated by oleic acid, palmitic acid, and fructose isomers, indicating the source of primary cooking emissions (To et al., 2000; Lyu et al., 2021; Huo et al., 2022). The second factor was regarded as vehicle emissions, due to the high loading of $NO_x$ (see discussions in section 3.1). Factor 3 with the dominance of levoglucosan signified biomass burning (Fabbri et al., 2009). Factor 4 was identified by 2-MT2 and cis-2-MBT, both of which were isoprene SOA tracers formed in $NO_x$-lean environments (Claeys et al., 2004). We define the fifth and sixth factor as ASOA-1 and ASOA-2 based on the highest percentage of phthalic acid and DHOPA, respectively, because phthalic acid and DHOPA have been shown to be the oxidation products of aromatics (Bunce et al., 1997; Jang et al., 1997; Kleindienst et al., 2004). Factor 7 was characterized by its high contribution to MA and moderate contribution to 2-HGA, both of which were OHDCA species. Our previous studies (Lyu et al., 2020; Huo et al., 2024) indicated that OHDCA in HK was attributed to aqueous photochemical formation rather than direct emissions, so this factor was termed as OHDCA-like SOA. Lastly, a high percentage of 9-oxononanoic acid was allocated to the eighth factor, which was deemed as OCOA (Huang et al., 2021; Lyu et al., 2021).

Further, the source contributions to $PM_1$-OM were determined. As shown in Figure 7a, they varied noticeably over time, underlining the value of source apportionment based on in-situ measurement of OA markers. Six sources explained a total of over 96% of the $PM_1$-OM. They were ASOA-2 ($23.6 \pm 7.1\%$, source contribution, the same below), PCOA ($19.8 \pm 3.5\%$), OHDCA-like SOA ($14.8 \pm 2.7\%$), vehicle emissions ($12.9 \pm 1.9\%$), ASOA-1 ($12.7 \pm 2.5\%$) and OCOA ($12.6 \pm 1.1\%$). Biomass burning and isoprene SOA (low $NO_x$) only accounted for $2.4 \pm 0.8\%$ and $1.1 \pm 0.3\%$ of $PM_1$-OM, respectively. The results demonstrated significant anthropogenic air pollution, either primary or secondary, contributing to $PM_1$-OM at this urban site. However, biogenic contributions might be somewhat underestimated, because i) isoprene SOA formed under high-$NO_x$ conditions or monoterpenes SOA were not identified, despite the inclusion of 2-MGA and HDMGA; and ii) OHDCA-like SOA could be formed through the oxidation of biogenic precursors (Hu et al., 2013; Huo et al., 2024). The dominant sources, especially PCOA and vehicle emissions, were different from those at the urban background site (Lyu et al., 2020), likely due to the depletion of primary emissions during the course of their airborne migration to the background site. It is noteworthy that the high contribution of ASOA-2 was mainly attributed to several samples in Case I and Case II, because the median contribution was much lower (4.5%), same for OHDCA-like SOA (6.5%). The following discussions based on different air mass origins reduce the impact of these samples on the overall mean.

In the most common scenario of marine air, PCOA ($28.3 \pm 4.8\%$) ranked the largest source of $PM_1$-OM, followed by vehicle emissions ($17.0 \pm 2.5\%$), OCOA ($14.8 \pm 1.3\%$), and ASOA-1 ($14.7 \pm 3.0\%$). As discussed in section 3.1, possibly due to the

enhancement of aqueous processes, some OA markers were elevated under the influence of the low-pressure troughs when backward trajectories still indicated marine air. Correspondingly, the contributions of ASOA-1 and OHDCA-like SOA to PM$_1$-OM increased to 20.9 ± 4.0% and 18.0 ± 2.3% during the trough periods, respectively, with the reduction in proportions of

365 primary sources. While the troughs appeared to increase the levels of 2-MTs (see section 3.2), the contribution of isoprene SOA (low NO$_x$) to PM$_1$-OM was still low (2.8%). In contrast, PCOA became the largest source of PM$_1$-OM (48.5 ± 9.0%) in the scenario of marine air without troughs, and in second and third place was OCOA (21.2%) and vehicle emissions (16.4%), respectively. Their total contribution of over 86% was striking but not necessarily unreasonable, because the marine air added little pollutants into the local air. Hence, OA pollution in this scenario exhibited predominantly local characteristics.

In Case I with continental air, ASOA-2 and OHDCA-like SOA with the contribution to PM$_1$-OM of 58.3 ± 21.4% and 15.8 ± 4.2% were the most predominant sources (Figure 7b). It is noteworthy that anthropogenic SOA, sum of ASOA-1, ASOA-2 and OCOA, accounted for 73.9 ± 21.3% of PM$_1$-OM. Except for OCOA, the enhancement of anthropogenic SOA in Case I was likely due to transboundary transport, given the increased concentrations of CO. Similarly, ASOA-2 and OHDCA-like SOA were also the leading sources in Case II with coastal air, contributing 48.3 ± 6.8% and 30.9 ± 2.4% to PM$_1$-OM,

respectively. Although the mass concentration of PM$_1$-OM attributed to OCOA in Case I (0.67 μg m$^{-3}$) and Case II (0.73 μg m$^{-3}$) was higher than in the marine air (0.47 μg m$^{-3}$), the percentage contributions were much lower (6.5% and 9.3% in Case I and Case II, respectively). Overall, the SOA contributions increased significantly with the rise in PM$_1$-OM concentration (Figure S11), consistent with the findings at a suburban site that PM$_{2.5}$ pollution events witnessed a notable rise in SOA tracer levels (Wang et al., 2022). However, it is noteworthy that PCOA made much less and even negligible contributions to the

PM$_1$-OM in the two cases. The reasons might be the same as those for the deficiency of PCOA markers in the non-marine air, as discussed in section 3.1.

Lastly, the diurnal variations of source contributions were also evident (Figure 7c). PCOA were responsible for up to 44.2 ± 7.8% of PM$_1$-OM during 18:00 – 20:00. Almost at the same time, biomass burning also contributed 2.4 ± 1.9% to PM$_1$-OM, although it was on average a small source. Therefore, short-term exposure to emissions of these sources might be a concern,

especially due to the fact that the highest levels of PAHs were related to biomass burning as discussed above. The total percentage contribution of SOA was relatively high and stable between 0:00 and 16:00, and decreased in the early evening as the cooking and biomass burning emissions intensified. The special diurnal pattern of total SOA contribution was a result of superposition of 5 SOA factors with different diurnal patterns, as shown in Figure S12. It is worth mentioning that the contribution of anthropogenic SOA could reach 54.3 ± 20.2% in the early afternoon (12:00 – 14:00).

**4 Conclusion**

As a ubiquitous composition of atmospheric fine PM, OA has been extensively studied. Nevertheless, it is still a major obstacle to further improving PM air quality in many places around the world, and also a meeting point in control of air pollution complex, such as PM$_{2.5}$ and O$_3$ in photochemical smog.

Taking advantage of an emerging technique for in-situ measurement of OA molecular markers, we reported by far the highest-resolution temporal variations of dozens of OA markers in a summer month in urban HK. The time-resolved data allowed to discuss the effects of transboundary transport and weather patterns (*e.g.,* troughs) on primary and secondary OA markers. DHOPA, phthalic acid, and isoprene SOA tracers were more enriched in the short-duration case with continental air, probably due to more abundant precursors and higher atmospheric oxidation capacity. The high RH brought on by troughs might have facilitated the formation of some OA species, such as OHDCA, phthalic acid and 2-MTs. Further, the diurnal variations of OA markers indicated emission patterns and SOA evolutions, such as the most intense cooking emissions in early evening and the subsequent OCOA formation. Regarding the sources of $PM_1$-OM, we uncovered the dominance of cooking and vehicle emissions in the scenario of marine air without troughs, which was the cleanest case. Even as a whole, primary cooking emissions explained ~44% of the $PM_1$-OM in early evening of this summer month. Anthropogenic SOA and OHDCA-like SOA were the main sources of $PM_1$-OM in the two cases with the intrusion of continental and coastal air. The SOA contributions, mainly from anthropogenic components, substantially increased when the PM air quality deteriorated. The findings demonstrated the need to properly manage local cooking and vehicle emissions and to control aromatics-derived SOA with concerted efforts on a regional or even superregional scale.

While this study provides valuable insights, we acknowledge there are several limitations. First, the molecular markers were not quantified, making it impossible to compare with the measurements elsewhere or compare among the detected species for absolute concentrations. It also hinders reuse of the data by other studies, such as concentration-based health risk assessment for some species. Second, some SOA tracers, such as those derived from monoterpenes and sesquiterpenes, were not fully characterized. This increased the uncertainty of OA source apportionment to some extent. Third, we reiterated the unexpected diurnal patterns of 2-MTs, but the analysis of the underlying reasons is speculative. Therefore, future work will be dedicated to improving the analytical performance of TAG and exploring complex mechanisms responsible for unique features of SOA tracers, including 2-MTs.

**Acknowledgements**

This study was supported by the Hong Kong Research Grants Council (RGC) via the National Natural Science Foundation of China/RGC joint research scheme (N_PolyU530/20), the National Natural Science Foundation of China (42061160478), and the General Research Fund (HKBU 15219621, HKBU 15209223). XL acknowledges the Tier 1 Research Start-up Grants provided by Hong Kong Baptist University (162912).

**Author contributions**

HL performed the study and drafted the manuscript. XL designed the study, oversaw interpretation of the results and revised the manuscript. YH, DY, and HL assisted in field campaign and processed relevant data. LX and HG led the project and revised the manuscript.

**Data availability**

All raw data are available upon request from the corresponding author Dr. Xiaopu Lyu.

**Competing interests**

The authors declare no competing interests.

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

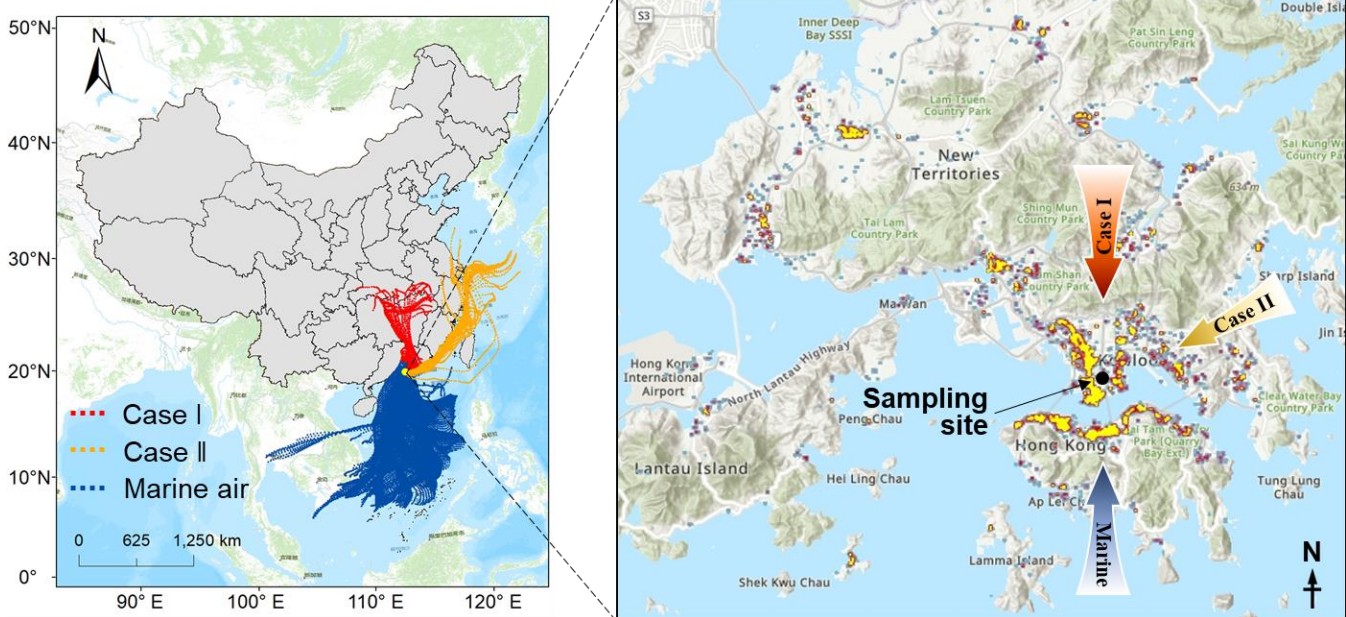

**Figure 1: Location of the sampling site with 72-h backward trajectories of 3 types of air masses and heatmap of restaurant distribution in HK.**

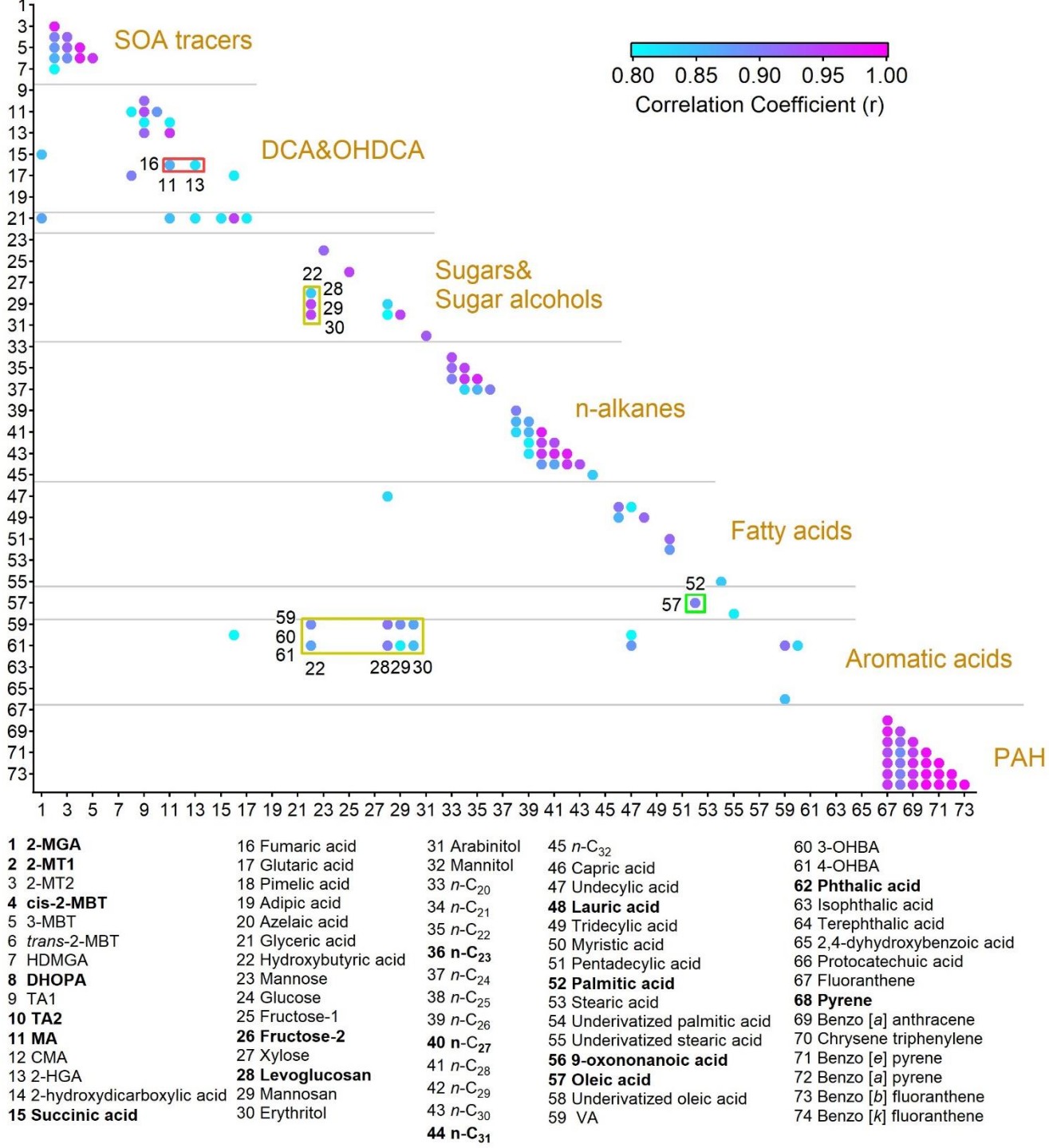

**1** **2-MGA**
**2** **2-MT1**
3 2-MT2
**4** **cis-2-MBT**
5 3-MBT
6 *trans*-2-MBT
7 HDMGA
**8** **DHOPA**
9 TA1
**10** **TA2**
**11** **MA**
CMA
2-HGA
2-hydroxydicarboxylic acid
**15** **Succinic acid**

Fumaric acid
Glutaric acid
Pimelic acid
Adipic acid
Azelaic acid
Glyceric acid
Hydroxybutyric acid
Mannose
Glucose
Fructose-1
**26** **Fructose-2**
Xylose
**28** **Levoglucosan**
Mannosan
Erythritol

Arabinitol
Mannitol
$n$-C$_{20}$
$n$-C$_{21}$
$n$-C$_{22}$
**36** **n-C$_{23}$**
$n$-C$_{24}$
$n$-C$_{25}$
$n$-C$_{26}$
**40** **n-C$_{27}$**
$n$-C$_{28}$
$n$-C$_{29}$
$n$-C$_{30}$
**44** **n-C$_{31}$**

$n$-C$_{32}$
Capric acid
Undecylic acid
**48** **Lauric acid**
Tridecylic acid
Myristic acid
Pentadecylic acid
**52** **Palmitic acid**
Stearic acid
Underivatized palmitic acid
Underivatized stearic acid
**56** **9-oxononanoic acid**
**57** **Oleic acid**
Underivatized oleic acid
VA

3-OHBA
4-OHBA
**62** **Phthalic acid**
Isophthalic acid
Terephthalic acid
2,4-dyhydroxybenzoic acid
Protocatechuic acid
Fluoranthene
**68** **Pyrene**
Benzo [*a*] anthracene
Chrysene triphenylene
Benzo [*e*] pyrene
Benzo [*a*] pyrene
Benzo [*b*] fluoranthene
Benzo [*k*] fluoranthene

**Figure 2: Pearson correlation coefficients between OA molecular markers with r not lower than 0.8. Bold are the representative species that we focus on in this study. Full names of the abbreviations are given in Table S3.**

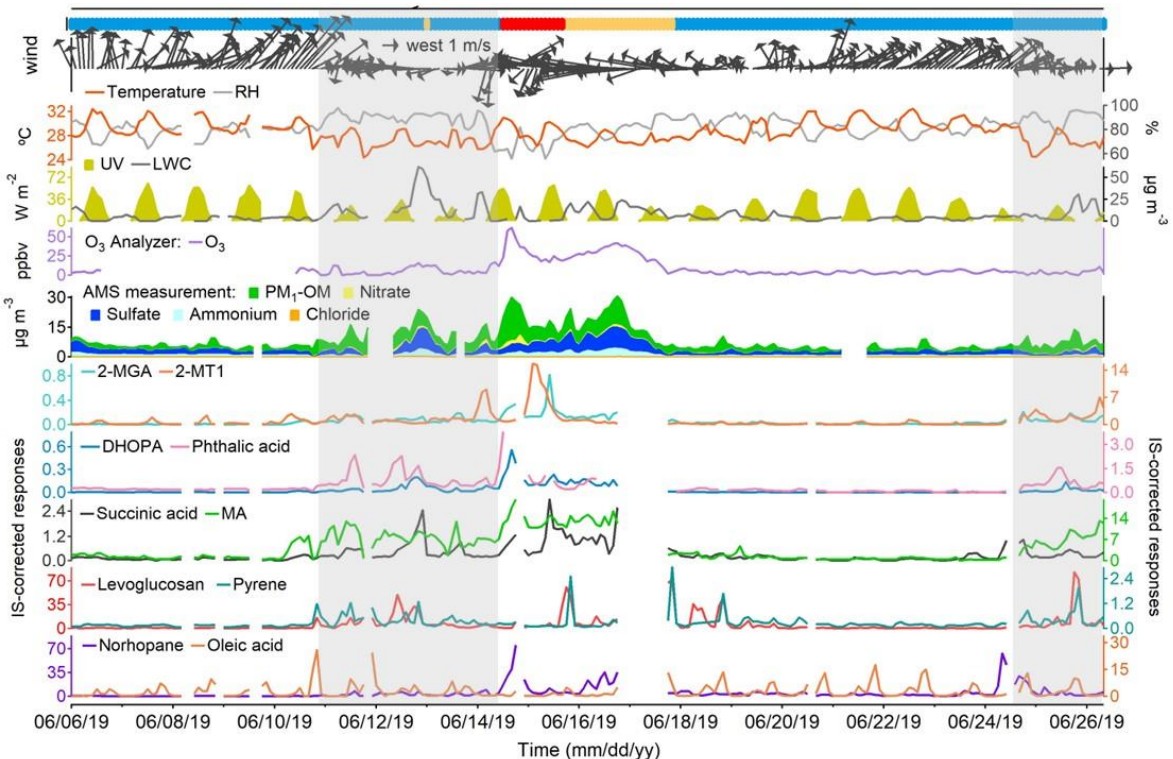

**Figure 3: Time series of representative OA markers, PM₁ compositions, trace gases, and meteorological conditions. Blue, orange, and red bars at the top represent marine, coastal, and continental air, respectively. Shaded areas represent the periods with troughs. Missing data are due to instrument maintenance.**

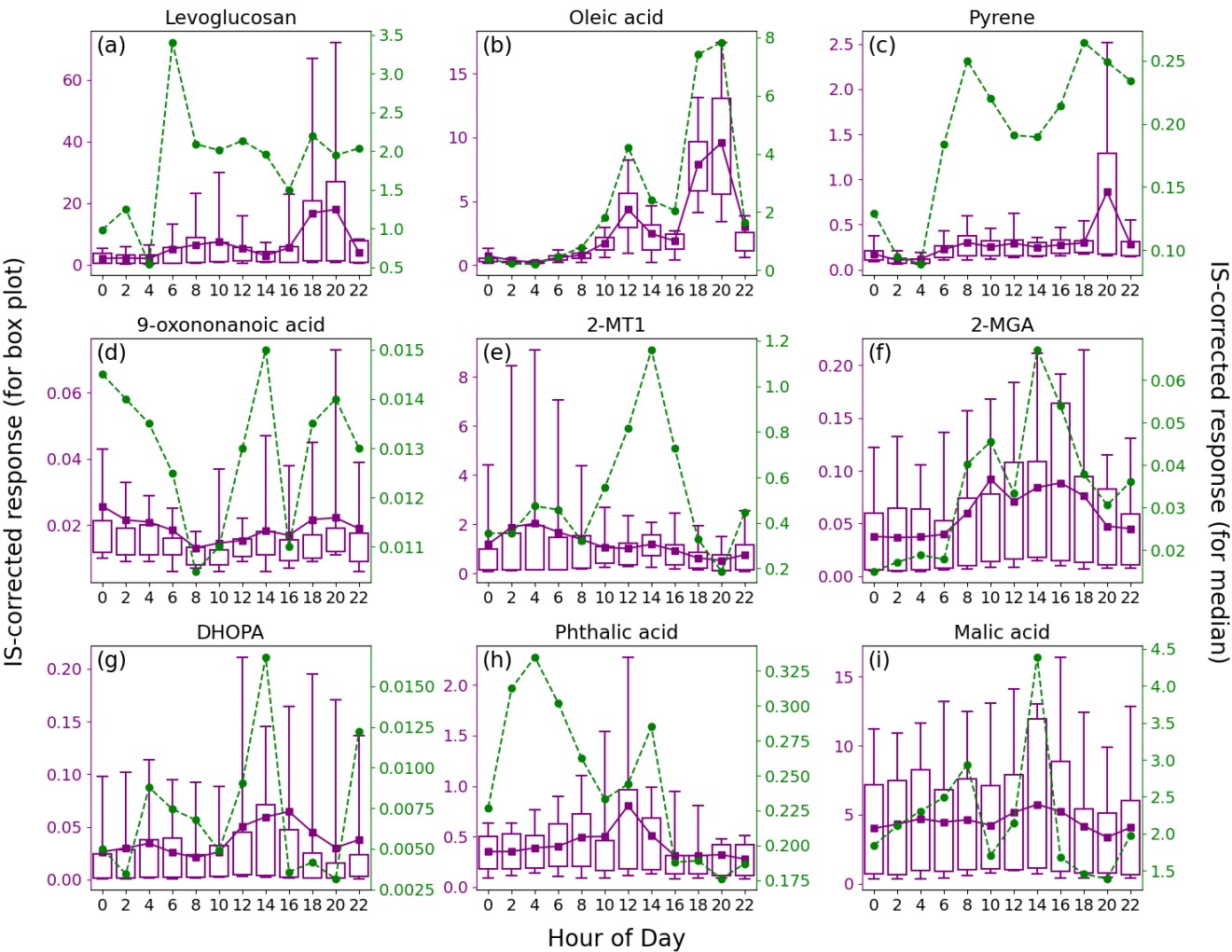

**Figure 4: Average diurnal patterns of nine representative OA markers. Tip of the top (bottom) whisker is the 95th (5th) percentile. Top and bottom of the box denote the 75th and 25th percentiles, respectively. The mean value is indicated by purple square, and green circle represents the median value.**

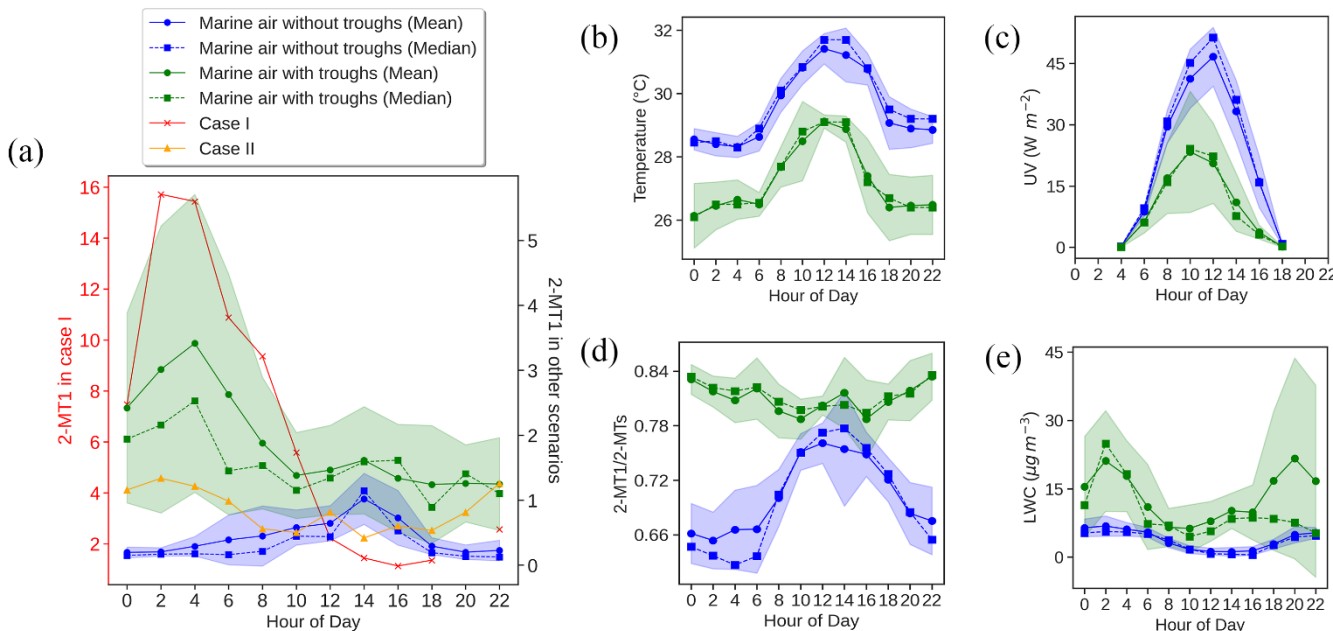

635

**Figure 5: Diurnal patterns of 2-MT1 in different scenarios (a) and the comparisons of temperature (b), UV (c), 2-MT1/2-MTs ratio (d), and LWC (e) between the marine air with and without troughs. Shaded areas represent 95% confidence intervals.**

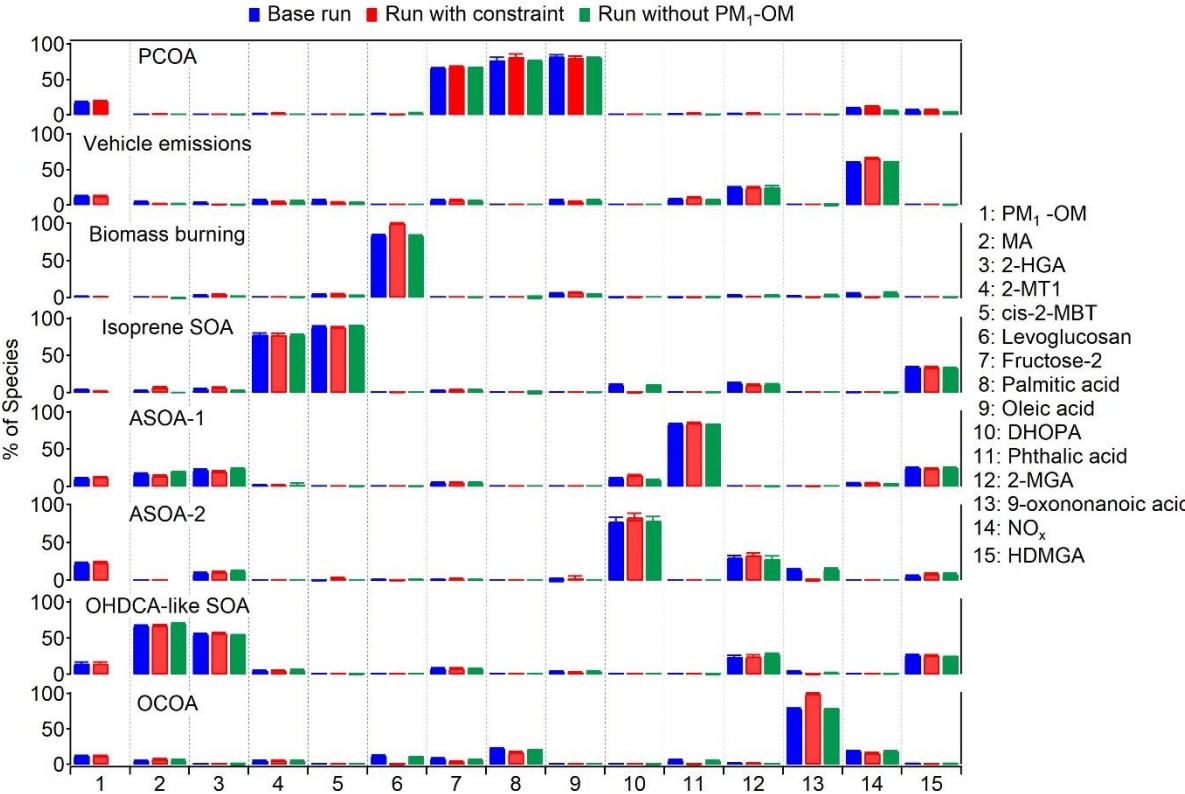

Figure 6: Average profiles of the eight factors resolved by PMF. The error bars represent 95% confidence intervals estimated by the BS method.

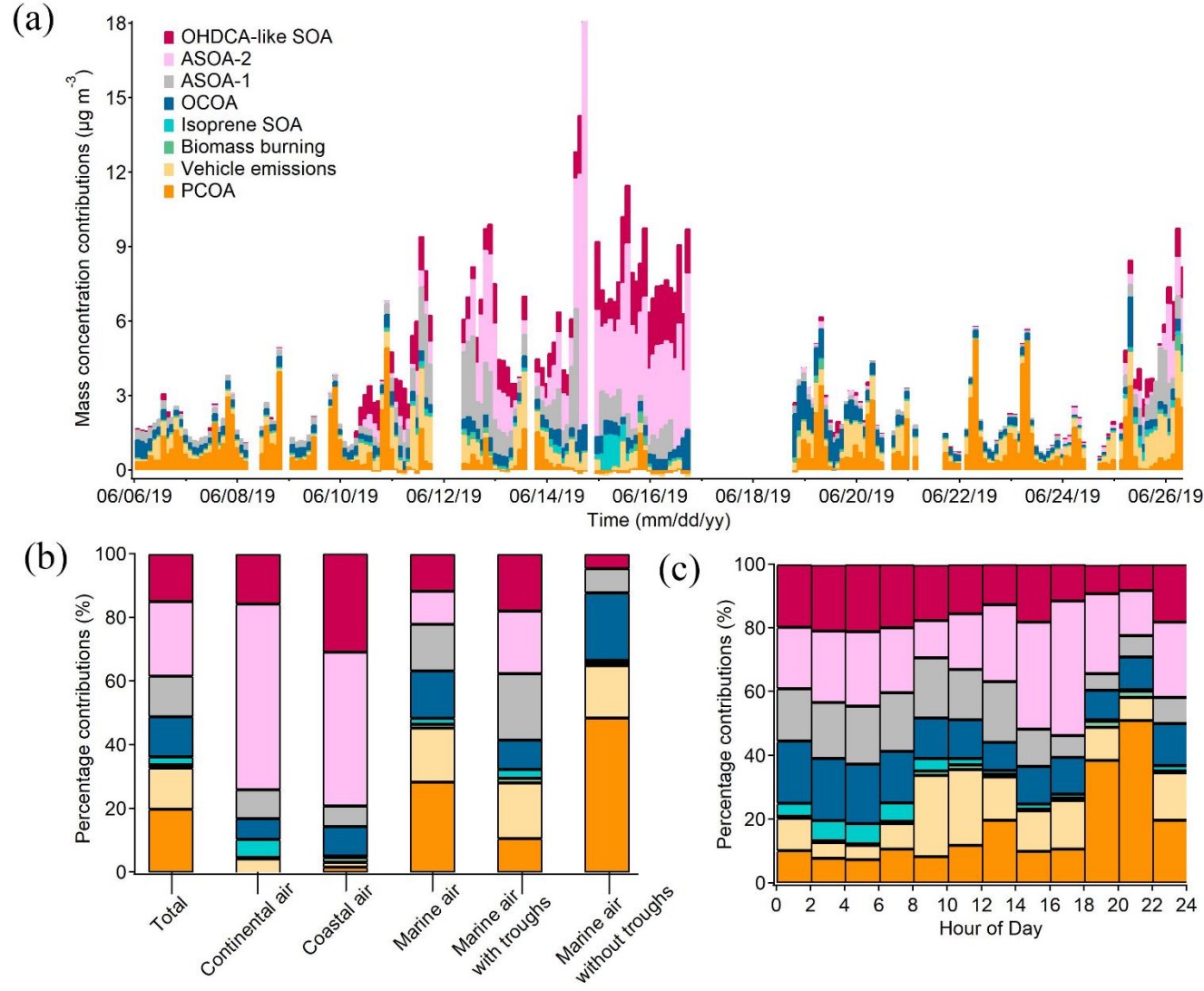

**Figure 7: Contributions of individual OA sources to PM₁-OM: time series of bihourly data (a); averages for the scenarios of different air masses (b); and diurnal variations (c).**