# Peer review of "In-situ measurement of organic aerosol molecular markers in urban Hong Kong during a summer period: temporal variations and source apportionment"

_EGUsphere, 2023_

## Author Comment (AC1)

**Response to reviewer #1**

This study characterized the diel dynamics of various organic markers in aerosols using a TAG system at urban Hong Kong during a summer period. The high time resolution observations of organic markers allowed to identify specific sources or processes that had crucial contributions to organic aerosols. This effort promoted our understandings of the sources of organic aerosols in urban HK, and the manuscript was well written. However, there have been some studies using TAG to apportion the sources of organic aerosols at urban/suburban areas. This study just looks like a supplement of TAG data in urban HK. Therefore, I recommend a major revision on the structure of writing/descriptions to point out some interesting findings. Detailed comments are as follows:

> *We are grateful for the review and constructive comments. Revisions have been made accordingly and are highlighted in blue throughout the text. For details, please see our point-by-point responses in italics below.*

**1.** The title looks inappropriate. First, the study was only conducted in a summer period less than one month. I feel the results could not represent "An overview of organic aerosols" in urban HK. Second, as stated by the authors, the concentrations of organic markers were not quantified, and some important markers such as monoterpene derived SOA tracer were not identified. In addition, nitroaromatics, which are key brown carbon species, were not detected. Nitroaromatics may also help evaluate the contributions of oxidation products from biomass burning and vehicle emissions to organic aerosols. Third, the title did not carry impressive information.

> *Thanks for the insightful comments on the title. With these points in mind, we have modified the title as follows.*

> **In-situ measurement of organic aerosol molecular markers in urban Hong Kong: temporal variations and source apportionment in summer**

**2.** The manuscript is mainly discussing the influence of different air masses (continental, coastal, and marine) on organic markers and OM sources. However, looking at Figure 3, there was only one day when air mass was from continental regions; only two days when air mass was from coastal regions and there was a one-day long maintenance of instruments when air mass was from coastal regions. One-day long observation is not representative. Therefore, grouping the data by continental, coastal, and marine air masses is not appropriate. For example, it is odd that cooking emissions had an insignificant contribution to OM when air mass was from continental region if the sampling site is surrounded by restaurants. This may be due to cooking plums did not significantly affect the sampling on just that day. It is feasible to regard the periods when continental and coastal winds were prevailing as cases to discuss the influences on organic markers but cannot regards the results as common situations when continental and coastal air mass dominated.

*Indeed, the number of days with continental or coastal air was too few to be representative. We treat them as special cases, and the related discussions are revised in the text.*

> However, the fraction of air masses originating from and passing over the mainland was very low (6%, red trajectories), same for those arrived at the site along the coastline of eastern and southern China (7%, orange trajectories). From a representative perspective, we discuss them as special cases, and the continental air and coastal air are labelled as Case I and Case II, respectively.

*For details, please refer to lines 87-90 and other places with Case I and Case II.*

*The low contributions of primary cooking emissions in the cases with continental and coastal air could be due to i) more conversion through oxidation, ii) uneven distribution of restaurants, and iii) sampling bias, which are discussed as follows.*

> However, the levels of PCOA markers were much lower in the cases with continental and coastal air than in the marine air. On one hand, this could be partially explained by the quicker chemical losses, due to the higher concentrations of oxidants, *e.g.,* $O_3$ (Table S3). Oleic acid with a carbon-carbon double bond in the molecule can be efficiently oxidized by $O_3$ and other oxidants (Zeng et al., 2020; Wang et al., 2021). The argument is supported by the fact that 9-oxononanoic acid, a typical OCOA tracer (Lyu et al., 2021), in the two cases was 1.47 – 2.00 times the level in the marine air. On the other hand, the differences could be related to the uneven distribution of restaurants on different air paths. As shown in Figure 1, the restaurant density in the areas where the continental and coastal airflows passed is substantially lower than south of the sampling site. Besides, it cannot be ruled out that the limited number of samples in the two cases coincidently did not capture the cooking emissions well.

*For details, please refer to lines 195-203.*

> Although the mass concentration of $PM_1$-OM attributed to OCOA in Case I (0.67 µg $m^{-3}$) and Case II (0.73 µg $m^{-3}$) was higher than in the marine air (0.47 µg $m^{-3}$), the percentage contributions were much lower (6.5% and 9.3% in Case I and Case II, respectively). Overall, the SOA contributions increased significantly with the rise in $PM_1$-OM concentration (Figure S10), consistent with the findings at a suburban site that $PM_{2.5}$ pollution events witnessed a notable rise in SOA tracer levels (Wang et al., 2022). However, it is noteworthy that PCOA made much less and even negligible contributions to the $PM_1$-OM in the two cases. The reasons might be the same as those for the deficiency of PCOA markers, as discussed in section 3.1.

*For details, please refer to lines 338-344.*

**3.** For the PMF analysis, it is not a good way to keep a factor which is unexplainable. The authors are encouraged to improve the PMF analysis by modifying inputs or others. As highlighted by the authors, cooking emissions should be a major source of OM at the

sampling site, while the contribution from cooking oxidation products to OM could not be evaluated. A previous study (Huang et al., Comparative Assessment of Cooking Emission Contributions to Urban Organic Aerosol Using Online Molecular Tracers and Aerosol Mass Spectrometry Measurements, Environ. Sci. Technol. 2021, 55, 14526-14535) used azelaic acid, nonanoic acid, and 9-oxononanoic acid to indicate the oxidation of cooking emissions. I noticed azelaic acid has been detected in this study (No. 20 in Figure 2). The authors may try to conduct an analysis to evaluate how significant of cooking oxidation is in contributing to OM mass.

*Excellent comments, especially regarding the aged cooking organic aerosol. We have made some adjustments based on the suggestions here and below and re-run the PMF model.*

*In this sampling campaign, both azelaic acid and 9-oxononanoic acid were detected by the TAG. Azelaic acid and oleic acid were similar in time series and diurnal patterns, therefore a moderate correlation between them ($R^2 = 0.57$), indicating that a considerable fraction of azelaic acid was derived from primary cooking emissions and/or quick formation from primary emissions. No similar result was found for 9-oxononanoic acid. However, the diurnal pattern of azelaic acid to oleic acid was quite consistent with that of 9-oxononanoic acid to oleic acid. In particular, the peaks several hours after the evening peak of oleic acid showed a clear evidence that primary cooking emissions experienced chemical aging. Based on these, we selected 9-oxononanoic acid as the tracer of aged cooking organic aerosol. In fact, Huang et al. (2021) also found that the loading of 9-oxononanoic acid in the factor of aged cooking organic aerosol was notably higher than that of the other tracers (e.g., azelaic acid, nonanoic acid).*

*To identify the factor of vehicle emissions, we used a hopane species as the tracer in the original manuscript. However, due to the lack of authentic standard, we do not have enough confidence in the identification of this hopane. Moreover, it did not show any correlation with $NO_x$ or any peak in the morning or evening rush hours. Hence, we have reservations about using this species. Instead, we adopt $NO_x$ as the tracer of vehicle emissions. First, the diurnal pattern of $NO_x$ was very similar to that observed at a roadside site 350 m away. Second, there was no correlation between $NO_x$ and levoglucosan, the biomass burning tracer, even in the hours with elevated levels of levoglucosan.*

*The new tracers, i.e., 9-oxononanoic acid, azelaic acid and $NO_x$, are discussed in before the source apportionment.*

*Lines 182-189:*

Although the marine air was supposed to be relatively clean, it contained elevated levels of $SO_2$ and $NO_x$, which correlated well with each other ($R^2 = 0.76$). This phenomenon was in line with that at the three roadside sites in HK, but was not identified at the general urban sites (Figure S3). Therefore, the close relationship between $SO_2$ and $NO_x$ on the days with marine air were likely attributable to local vehicle emissions rather than ship emissions. For all the air masses, we also note that

the variation of NOx throughout the day was highly consistent with that observed at a nearby roadside site (Figure S4). Moreover, there was no correlation between NOx and levoglucosan, even during the period with elevated levels of levoglucosan, eliminating the likelihood of high biomass burning contribution. Therefore, NOx is regarded as a tracer of vehicle emissions in this study.

*Lines 244-247:*

Oleic acid was obviously enhanced at noon and in early evening, when intensive cooking activities took place in restaurants and at home, same for fatty acids, azelaic acid, and fructoses (Figure S7). The consistency of azelaic acid with oleic acid ($R^2 = 0.57$) indicated that a considerable fraction of the azelaic acid was derived from or quickly formed in primary cooking emissions.

*Lines 253-257:*

As a tracer of OCOA, 9-oxononanoic acid got higher levels at night. This was in line with the observations in an indoor air quality study (Lyu et al., 2021), and was due to the $O_3$-initiated oxidation of cooking emissions that peaked in the early evening. Although the diurnal pattern of 9-oxononanoic acid was totally different from that of oleic acid and azelaic acid, the variation in the ratio of 9-oxononanoic acid to oleic acid was consistent with that in the azelaic acid to oleic acid ratio (Figure S9), both reflecting the chemical aging of cooking emissions.

*With these changes, there is no unexplainable factor, and we got a factor representing oxygenated cooking organic aerosol (OCOA). Please refer to section 3.3 for details of the revisions.*

**4.** It was interesting to identify the periods lasting a few days when trough played a role. The meteorological parameters such as RH, Temperature, solar radiation, wind speed showed significant differences between "trough" days and "non-trough" days. I think it is valuable to focus on discussing the variations of organic tracers, especially SOA tracers, during the two distinct periods. It is good to see that in the manuscript the authors have pointed out the increases of phthalic acid and DCAs during "trough" days, indicating an aqueous formation of the species. The authors are encouraged to find more markers that had significant difference between the periods. Hopefully, the authors can evaluate how significant of aqueous formation is in contributing to OM mass. This would be a very interesting part.

*Thanks. In fact, a wide range of POA and SOA tracers experienced an increase on the 'trough' days. We discussed some of the increases in different parts of the manuscript, such as nocturnal enhancement of 2-MT (section 3.2), positive response of phthalic acid to the product of radiation and relative humidity (section 3.3), and potentially aqueous photochemical formation of OHDCA (section 3.3). The revised manuscript integrates these discussions and includes some additional discussions on other species. However, the*

*discussions on aqueous processes are speculative, and it is hard to quantify the contributions to PM$_1$-OM.*

> In fact, the troughs witnessed enhancements of a wide range of species relative to their levels in the marine air without troughs. The highest change ratio for the POA markers was 5.3 for benzo[a]pyrene. However, more significant increases were observed for many SOA markers, including DHOPA, C$_5$-alkenetriols, OHDCA species, and phthalic acid, implying the role of processes other than transport and atmospheric mixing. It is noteworthy that the troughs featured lower temperature and much weaker radiation, while the RH and LWC were higher. A recent study demonstrated the aqueous photochemical formation of OHDCA in HK (Huo et al., 2024). While it is unknown whether this was also responsible for the increases of the other SOA markers, a moderate correlation (R$^2$ = 0.41) was identified between phthalic acid and the product of RH and UV during the trough periods and was not found at other times (Figure S6). Besides, the rise in C$_5$-alkenetriols resembled the changes in 2-MTs, which mainly occurred at night and could also be linked to aqueous processes, as discussed below. Therefore, the high RH and LWC might be one of the leading factors, if not all, of the SOA enhancements in the presence of troughs.

*For details, please refer to lines 224-234.*

**5.** Some inconsistencies may exist. For example, in Figure 7a, cooking emissions (dark yellow) constituted a significant part of OM sources during June 15th to 17th (continental and coastal air masses dominated according to Figure 3), while in Figure 7b and text, cooking emission was a very minor source of OM during the period. Another example is the X axis of Figure S8 shows a OM range of 0-22 μg/m$^3$ while the Y axis of Figure 7a only reached 15 μg/m$^3$. If X axis of Figure S8 shows the observation result, then there should be an unresolved percentage in Figure S8.

*We sincerely appreciate the careful review. The errors have been corrected. Please refer to the updated Figure 7, Figure S10 and discussions in the text.*

**6.** line 185-188: the explanation may need to be modified. Just like I mentioned above, one-day observation for continental air mass dominated period may not capture the influence from cooking emission. During fall-winter period, most air masses may come from the north, and I guess you may still find the contribution from cooking emissions if a long period of observation is available.

*As stated above, the reasons for lower levels of cooking emission tracers in the two cases with continental and coastal air could be multifaceted. The explanation has been revised, and please refer to our response to comment #2.*

**7.** line 227: why not examine the correlation between pyrene and hopanes? $NO_x$ can also be emitted from biomass burning.

*We tried but did not see any correlation between them. In fact, this comment reminded us to revisit the hopane data. As stated in the response to comment #3, we do not have enough confidence in the species identification and data quality. Instead, NOx seemed to be a better indicator of vehicle emissions.*

**8.** line 290: add "respectively" after $PM_1$-OM.

*Accepted with thanks.*

**9.** line 318: please try to add azelaic acid in PMF analysis to see if cooking oxidation factor can be resolved.

*As stated above, we use 9-oxononanoic acid as the tracer of OCOA. Please see our response to comment #3 for details.*

**10.** Figure 1. How about show the locations of restaurants in the map?

*Accepted. We show a heatmap of restaurant distribution in HK. Please see the updated Figure 1.*

**11.** Figure 3. Please add the time series of DCAs.

*Thanks for the suggestion. We have added the time series of succinic acid, a representative DCA species we measured. Its variations are also briefly discussed. Please see the updated Figure 3.*

---

## Author Comment (AC2)

**Response to reviewer #2**

Comments from reviewer #2:

Major concerns

1. The study determines a series of organic markers in PM2.5 with two-hour resolution in Hong Kong for one month. However, the data management is inadequate. The mean and median values for most species are very different (at least for those shown in Figure 4), probably due to variability in compound concentrations. Therefore, it is required to know the distribution of the variables, to confirm the assumption of the normal distribution done by the authors.

2. The authors must show the distribution of the variables, thereby justifying the type of statistics to be used, whether parametric or non-parametric, or some other mathematical proposal for data management. They should double-check any statements or suggestions they made.

3. Throughout the document, no statistical evidence is shown to support the conclusions of the values comparisons between the different periods. Nor the use of Pearson correlations. Subjectivity should be avoided in the discussion. Appropriate statistical tests will verify the hypotheses proposed by the authors. The article cannot be published in its current state and must be resubmitted for consideration.

*We sincerely appreciate the professional comments on data management. Since the three major concerns are related to each other, we provide a combined response to them. In brief, we have examined the distributions of the observational data with Shapiro-Wilk tests and found that in most cases they did not follow a normal distribution, which is actually a common phenomenon for urban air quality measurement. As a response, we show more metrics of the data, including mean±95% confidence interval, median, 25$^{th}$ percentile, and 75$^{th}$ percentile. Moreover, non-parametric Mann-Whitney U tests have been performed to compare any two sets of data that does not conform to the normal distribution. We've also changed all the coefficients of determination to Pearson correlation coefficients throughout the manuscript. The methods are described in a new section 2.4.*

**2.4 Statistical analysis**

Due to uneven emission patterns, non-linear chemistry and changing meteorological impacts, the distribution of urban air pollutants concentration is often uncertain. This determines how we describe the data. In this study, we examined the data distribution with the Shapiro-Wilk tests. It was found that the concentrations of most species studied did not follow a normal distribution, neither over the entire observation period nor within sub-periods defined by air mass categories, with only a few exceptions (e.g., palmitic acid in Case I and 2-MGA in Case II). Therefore, it is insufficient to analyze and discuss the data based only on the mean values. As a supplement, we also show the 25$^{th}$ percentile, median

and 75[th] percentile where necessary. However, to avoid redundancy, not all the metrics are referenced in all discussions. Moreover, given the non-normal distribution of the data, non-parametric Mann-Whitney U tests were performed to compare any two sets of data. The significance levels are expressed as $p$ values. Both the Shapiro-Wilk test and the Mann-Whitney U test were implemented using the scipy.stats package in Python. All the correlation coefficients shown in this paper are derived from Pearson correlation analysis and are significant ($p < 0.05$), unless otherwise specified.

*For details, please refer to lines 178-189.*

*Correspondingly, the discussions have been updated with the addition of median where necessary. It is our pleasure to invite the reviewer to review the revisions highlighted in blue throughout the manuscript and supplement. In this file, our point-by-point responses are formatted in italics.*

Minor comments

1. Lines 100-104. Place the mass of the compound isotopically labeled with carbon 13, added to the TAG.

*The volume of internal standards that were added on top of the samples was 5.22 μL and the concentrations for 41 species were in the range of 0.0625–0.5 ng/μL. So, the added mass varied between 0.33 ng and 2.61 ng for different species. They were either deuterated ($^2$H-containing) or $^{13}$C-containing compounds, which had little impact on the target compounds in the air. More information of the internal standards is provided.*

> A 5.22 μL of mixture of deuterated ($^2$H-containing) or $^{13}$C-containing compounds at the concentration of 0.0625-0.5 ng/μL was used as internal standards (IS), which was injected on top of every ambient sample and subject to the same treatment and analysis procedures. Therefore, the ISs could track and correct for the changes in instrument sensitivity. More details about the AMS and TAG operations, including the IS information, can be found in our previous studies (Lyu et al., 2020; Huo et al., 2022).

*For details, please refer to lines 109-113.*

2. Lines 105-109. Change below to "throughout the document".

*Accepted with thanks.*

3. Lines 110-114. This way of associating the variables with their respective concentrations is confusing. Each value must be associated with the corresponding contaminant. Apply to the entire document.

*Thanks for the suggestion. In view of the other comments on data management, we are committed to showing more metrics to describe the data. Thus, the values are removed here and presented in a supplementary table.*

Table S2 shows the various metrics (mean, median and percentiles) of the trace gases, meteorological parameters and LWC.

*For details, please refer to lines 121-122 and Table S2.*

4. Lines 120-129. The description is very confusing. Explain more clearly why organic contaminants were not quantified with the TAG. Explain what the authors refer to as "internal standard scaled peak areas". Check if it is an accepted term in the scientific community. Enter the corresponding equation for calculation.

*As stated above, the fixed amount of internal standards was injected on top of every ambient sample and subject to the same subsequent processes, e.g., derivatization, transfer and analysis. Therefore, the variations of their signals over time were caused by changes in instrument sensitivity only. For example, the sensitivity decreases when the ion source of the mass spectrum gets dirty. The unstable sensitivity hinders understanding of the true variations of target compounds in the air. It is why we need to use internal standards to correct for the measured variations in target compounds. Specifically, we calculated the ratio of the peak area of target compounds to the peak area of the corresponding internal standards, which was defined as IS-scaled peak area. This method is widely used in chemical analysis (Williams et al., 2007; Kreisberg et al., 2009; Williams et al., 2010). However, inspired by this comment, we think it is more appropriate to define the ratio as IS-corrected response. The corrected responses would be converted to concentrations of target compounds if we had external standards to establish the concentration-response relationships. Unfortunately, we ran out of external standards in this sampling campaign, thus the target compounds were not quantified for their concentrations. The description has been revised.*

Given the consistent amount of ISs injected on top of all individual samples, the variations in peak areas of ISs over time could only be attributed to changes in instrument sensitivity (including recovery rate). To understand the true variations of target compounds in the air, we calculated the ratio of the peak areas of target compounds to the peak areas of corresponding ISs with similar structures and close retention time, and the ratio is deemed as IS-corrected response. The pairing of target compounds and ISs are the same as that adopted in Lyu et al. (2020). However, we ran out of the external standards (multiple concentrations of authentic standards that are exactly the same as target compounds or surrogate standards) in this sampling campaign. Therefore, the relationship between IS-corrected response and concentration was not determined, and the concentrations of the detected species were not quantified. We use the IS-corrected responses of target compounds for analysis instead.

*For details, please refer to lines 130-138.*

5. Lines 151-155. Explain in detail and in the supplementary material the calculation of the method detection limit (MDL). Explain why it is only one value. In general, it should be one MDL for each compound. What sense does the MDL in µg/m$^3$ if the units of the compounds are shown in relative areas?

*The MDL of 0.201 µg/m$^3$ was for PM$_1$-OM, an individual component quantitatively measured by AMS. The calculation method is described in a supplementary text. It was the molecular markers detected by TAG that were not quantified. Their MDLs were not available, and we adopted those determined in a previous study.*

> The MDL for PM$_1$-OM measured by AMS was determined to be 0.201 µg/m$^3$, as described in Text S1. Since the molecular markers measured by TAG were not quantified in this sampling campaign, the MDLs determined for the same instrument ~8 months earlier were adopted (Lyu et al., 2020). For the newly detected species, *e.g.,* phthalic acid, 2-MGA, cis-2-MBT, and 9-oxononanoic acid, we assume 10% of the average IS-corrected response as the dimensionless MDL.

For details, please refer to lines 164-168 and Text S1.

6. Lines 170-174. Avoid subjectivity throughout the document. It is not appropriate to simply say: slightly higher, slightly lower, etc., because nothing conclusive is got it. The level of significance associated with the statistical test must be set. In this case the averages are compared. As mentioned above, the use of the average must first be justified, and subsequently, the result of the comparison test and the significance level must be given. This applies to the rest of the comparisons made throughout the document.

*We appreciate the reviewer for the scientific rigor. In response, we have examined the distribution of the data with Shapiro-Wilk test, used more metrics (mean, median and percentiles) to describe the data, and conducted non-parametric Mann-Whitney U tests to compare two sets of data that does not follow normal distribution.*

> Table S4 lists the statistics of the PM$_1$-OM, sulfate, nitrate, and ammonium in PM$_1$. PM$_1$-OM was found to be the most abundant component, followed by sulfate, ammonium and nitrate, whether based on the mean or median. The fraction of PM$_1$-OM (47.1 ± 2.2%) was comparable to that at an urban background site in HK (43.8%), but significantly lower ($p < 0.05$) than that at a roadside site ~350 m away (57.7%) (Yao et al., 2021,2022). From the perspective of air mass categories, PM$_1$-OM was the lowest in the most common marine air, followed by that in Case II with coastal air and Case I with continental air, same for CO and O$_3$ (Tables S5-S7).

*For details, please refer to lines 193-198.*

*As stated above, With the improvement of data management, many discussions are updated, especially in section 3.2. However, we still retain the discussions based on averages, because the small number of high values are true values and reflect the short-duration but intensive emissions or chemical formation. Please review the detailed revisions in the revised manuscript.*

7. In a separate section, the statistical tests must be described.

*As stated above, we have improved the data management based on the insightful comments. For details, please refer to our response to the major concerns and the new section 2.4.*

8. Lines 190-194. This assertion is not clear. Figure S4a shows that levoglucosan maximum indeed comes from the east, but between 7 and 18 km away from the site in the hour before its emission. From figure 1, it is not clear how the authors assume that the biomass combustion is located in the first kilometer east of the site. Clarify.

*The original Figure S4a (Figure S5a in this revised supplement) shows higher levels of levoglucosan under easterly winds. However, it does not indicate the distance between the emission source and the sampling site, because we do not know the time between levoglucosan emission and detection. Besides, the TAG did not analyze instant samples but collected the sample for 90 minutes and analyzed it afterwards in each 2h cycle. We extrapolated the local origin of levoglucosan based on the following facts: i) there were several funeral parlors and temples east of the sampling site within 1km, where burning incense, candles, banknotes, paper products was common; and ii) there was no known source of biomass burning further east. The funeral parlors and temples were marked in the original Figure 1, which are replaced with the distribution of restaurants – a much larger source of $PM_1$-OM – in the updated Figure 1. For clarity, the discussions are revised as follows.*

> Further, we found that high levels of levoglucosan were associated with easterly winds (Figure S5). As stated in section 2.1, the burning of incense, candles, banknotes, paper products was common within 1 km east of the site. There was no known source of biomass burning further east, and east of HK are waters. Therefore, the observed variations of levoglucosan were mainly attributed to the combined effect of local emissions and wind patterns, different from the regional transport of biomass burning plumes from mainland to HK in cool seasons (Sang et al., 2011; Chow et al., 2015).

*For details, please refer to lines 223-227.*

9. Lines 215-219. What is the evidence to associate these alkanes with sacrificial activities? Explain what the authors refer to as sacrificial activities.

*With 'sacrificial activities', we meant burning of incenses, candles, joss papers and paper artifacts nearby. Studies identified alkane emissions from burning some of these biomass products, and the observed alkanes exhibited similar diurnal patterns to biomass burning tracers (e.g., levoglucosan). That's why we attributed the alkanes partially to 'sacrificial activities'. For clarity, the following revisions are made.*

> Besides, both the bihourly mean and median $C_{26}$ – $C_{31}$ *n*-alkanes were observed with obvious peaks at 8:00 and 20:00 and might also be related to the burning of incenses, candles, joss papers and paper artifacts nearby, which was confirmed in other scenes (Lyu et al., 2021; Song et al., 2023). Other sources, *e.g.,* vehicle emissions, might also contribute to the peak at 8:00 that was more pronounced than levoglucosan.

*For details, please refer to lines 266-270.*

10. Lines 220-224. Azelaic acid is not shown in Figure S5.

*Azelaic acid is shown in the updated Figure S7. In response to the other reviewer, more discussions on azelaic acid are added. Please see lines 270-273 and lines 283-285.*

11. Lines 220-229. The pyrene is a clear example that the average is not the best metric to compare periods. In this case, the median of the hour 20:00 is like the rest of the hourly medians. This change the discussion and conclusions.

*Indeed, mean-based discussion may not be enough for the data that does not follow normal distribution. As stated above, we adopt more metrics in the revised manuscript, and the discussion is revised as follows.*

> Pyrene, as a representative of PAHs, was observed with higher median and mean values during 6:00-22:00. The median values showed peaks in the morning and early evening, while the mean value was the highest at 20:00. Interestingly, the high levels of pyrene correlated well with levoglucosan (r = 0.94) at 20:00 (Figure S9) and exhibited no correlation with oleic acid which also had the highest levels in the early evening. Therefore, the burning of biomass products nearby was responsible for the occasionally elevated levels of PAHs in the early evening. At the other times, the variation of pyrene was consistent with that of $NO_x$ (r = 0.80), indicating their common source of vehicle emissions. This also explained the bimodal distribution of the bihourly median values of pyrene.

*For details, please refer to lines 273-279.*

12. Line 227. It is well known that pyrene and $NO_x$ are emitted by vehicles that burn diesel. Could this correlation be associated with this source?

*Yes. Pyrene at times other than 20:00 was likely to be emitted by vehicles. Please see our response to comment #11.*

13. Line 228. Does the term funerary refer to crematoriums for humans and animals? since there are funeral homes that do not have this service. If so, the authors should cite studies showing the type of PAHs emitted by these sources found in this study. This information will be useful to strengthen their hypothesis.

*Crematoriums for humans or animals were not the service of the nearby funeral parlors. The high levels of PAHs at 20:00 were likely related to the burning of biomass products (incenses, candles, joss papers and paper artifacts). In Chinese culture, burning these things to gods and souls is a kind of memorial and offering. There have been many studies showing PAH emissions from biomass burning.*

*However, we do not want to challenge the culture with this little piece of evidence. And for clarity, we rephrase this sentence to highlight the burning of biomass products.*

> Therefore, the burning of biomass products nearby was responsible for the occasionally elevated levels of PAHs in the early evening.

*For details, please refer to lines 276-277.*

14. Line 240. It's confusing. Is it sum or division?

*The sentence has been rephrased for clarity.*

> According to Noziere et al. (2011), the ratio of 2-methylerythritol to the sum of 2-methylerythritol and 2-methylthreitol, referred to as 2-MT1/2-MTs hereinafter, indicates the sources of 2-MTs: primary emissions (0.35), $NO_x$-lean photooxidation (0.61), $NO_x$-rich photooxidation (0.76), and aqueous phase oxidation (0.90).

*For details, please refer to lines 297-299.*

15. Figure 3. Why does the time scale go from 6 to 26? What are the units? Eliminate the $\mu g/m^3$ on the scale on the right. Change color between ammonium, nitrate and coastal air bar. It's confusing. Change the color between the UV and continental air bar. It's confusing.

*The number on the x-axis represented date in June 2019, which is updated with a standard date format. The colors are also changed to avoid confusion. Other changes can be found in the updated Figure 3. The unit $\mu g/m^3$ is for the species measured by AMS, which were quantified. We have added a note into this panel highlighting they were measured by AMS.*

16. Figure 7a. Why does the time scale go from 6 to 26? What are the units? Improve sharpness.

*Date format changed. The figure has been updated as suggested.*

17. Figure S2. Why does the time scale go from 6 to 26? What are the units? Place the color bar for marine, continental and coastal air, as in figure 3.

*Date format changed. The figure has been updated as suggested.*

18. Figure S5. Explain the calculation to normalize the variables.

*The normalization method is described in a supplementary text, which is referenced in the figure caption.*

**Figure S7.** Diurnal patterns of selected OA markers represented by the normalized mean values of IS-corrected response (see Text S2 for normalization method). The species are grouped based on the similarity in patterns.

**Figure S8.** Same as Figure S7, but showing normalized median values (see Text S2 for normalization method).

**Text S2. Normalization of diurnal profiles**

The analysis of diurnal profiles focuses on the variations rather than magnitudes of the species. To accommodate the species with same patterns but different orders of magnitude in a same figure, we adopt a linear normalization approach to process the data, according to the formula:

$$Normalized\ value_t = \frac{x_t - min}{max - min}$$

where $x_t$ is the mean (or median) value of the species in hour $t$; and $min$ and $max$ are the minimum and maximum of all the mean (or median) values throughout the day, respectively.

*For details, please refer to Text S2 and the updated Figures S7-S8.*

19. Figure S6. There is inconsistency between the graphs and the legend.

*Thanks for pointing out this mistake. The figure is updated.*

20. Table S2. Verify the names of the compounds.

*Verified. Thanks.*

*References:*

*Kreisberg, N. M, Hering, S. V., Williams, B. J., Worton, D. R., and Goldstein, A. H.: Quantitation of hourly speciated organic compounds in atmospheric aerosols, Measured by in-situ thermal desorption aerosol gas chromatography (TAG), Aerosol Sci. Technol., 43(1), 38–52, 2009.*

*Williams, B. J., Goldstein, A. H., Millet, D. B., Holzinger, R., Kreisberg, N. M., Hering, S. V., White, A. B., Worsnop, D. R., Allan, J. D., and Jimenez, J. L.: Chemical speciation of organic aerosol during the International Consortium for Atmospheric Research on Transport and Transformation 2004: Results from in situ measurements, J. Geophys. Res., 112, D10S26, doi:10.1029/2006JD007601, 2007.*

*Williams, B. J., Goldstein, A. H., Kreisberg, N. M., Hering, S. V., Worsnop, D. R., Ulbrich, I. M., Docherty, K. S., and Jimenez, J. L.: Major components of atmospheric organic aerosol in southern California as determined by hourly measurements of source marker compounds, Atmos. Chem. Phys., 10, 11577–11603, https://doi.org/10.5194/acp-10-11577-2010, 2010.*

---

## Referee Report (RR1)

The reviewer appreciates the authors' efforts in addressing the previous comments and criticisms associated with the submitted manuscript. The revised manuscript has improved and is recommended for publication in ACP, subject to the incorporation of the following minor suggestions aimed at enhancing clarity and consistency:

Lines 20 & 22: It appears that the terms "scenarios" and "cases" are used interchangeably. To prevent any confusion, please choose one term and consistently apply it throughout the manuscript.

Line 21: Suggest replacing "sources" with "source components" to more accurately reflect that PMF factors may represent not only direct emission sources but also formation processes. This terminology will better indicate that these factors are mathematical constructs that help in understanding the contributions to the data, including combinations of multiple sources or transformation processes, rather than specific, isolated sources.

Line 21: The term "unambiguous" is used in the abstract. If this pertains to the specificity enabled by the TAG method described in the main text, please clarify this connection directly in the abstract to enhance reader understanding.

Lines 323 & 326: To avoid potential confusion (and follow standard language in the PMF community), replace "simulations" with "modelling" when referring to PMF runs. For instance, use "…between the PMF-modelled and observed…" in line 326 and apply this change consistently throughout the manuscript and in Table S8 caption.

Line 391: Change "composition" to "component".

Lines 396 & 397: The terms "markers" and "tracers" seem to be used interchangeably. Please select one and use it consistently throughout the document. Additionally, provide a justification if these terms have distinct definitions.

---

## Author Response (AR2)

**Response to the Editor**

The authors have addressed the major comments from the referees to the first version of the manuscript. I ask them to take into account the minor comments to the revised version.

*Thanks. The manuscript has been further revised based on the minor comments of the reviewers. Our responses and revisions are in **italics** and highlighted in **blue**, respectively.*

**Response to the Referee #1**

The authors have properly addressed the questions and comments raised by the reviewers. I have no further comments except one suggestion: Please revise the title of manuscript as "In-situ measurement of organic aerosol molecular markers in urban Hong Kong during a summer period: temporal variations and source apportionment".

*Accepted with thanks.*

**Response to the Referee #2**

The comments have been well addressed. The paper is ready for publication.

*Thanks.*

**Response to the Referee #3**

The reviewer appreciates the authors' efforts in addressing the previous comments and criticisms associated with the submitted manuscript. The revised manuscript has improved and is recommended for publication in ACP, subject to the incorporation of the following minor suggestions aimed at enhancing clarity and consistency.

*Thanks. We have made further revisions based on the minor comments. Please see our item-by-item responses below.*

1. Lines 20 & 22: It appears that the terms "scenarios" and "cases" are used interchangeably. To prevent any confusion, please choose one term and consistently apply it throughout the manuscript.

*The term 'scenario' has been replaced with 'case' throughout the manuscript.*

2. Line 21: Suggest replacing "sources" with "source components" to more accurately reflect that PMF factors may represent not only direct emission sources but also formation processes. This terminology will better indicate that these factors are mathematical constructs that help in understanding the contributions to the data, including combinations of multiple sources or transformation processes, rather than specific, isolated sources.

*Thoughtful comment. However, the term 'source components' may confuse the readers, because it can also refer to chemical components of sources. We add an annotation to the 'source' in the methodology section.*

Here, a 'source' refers not only to a specific and isolated source of direct emissions, but also to a combination of multiple sources (e.g., vehicle emissions) and/or transformation processes (e.g., SOA).

*For details, please see lines 163-164.*

3. Line 21: The term "unambiguous" is used in the abstract. If this pertains to the specificity enabled by the TAG method described in the main text, please clarify this connection directly in the abstract to enhance reader understanding.

*We delete the term 'unambiguous' to avoid overstatement.*

4. Lines 323 & 326: To avoid potential confusion (and follow standard language in the PMF community), replace "simulations" with "modelling" when referring to PMF runs. For instance, use "…between the PMF-modelled and observed…" in line 326 and apply this change consistently throughout the manuscript and in Table S8 caption.

*Accepted with thanks.*

5. Line 391: Change "composition" to "component".

*Accepted with thanks.*

6. Lines 396 & 397: The terms "markers" and "tracers" seem to be used interchangeably. Please select one and use it consistently throughout the document. Additionally, provide a justification if these terms have distinct definitions.

*For consistency, we replaced 'tracer(s)' with 'marker(s)' throughout the main text.*